

# Implicit-explicit (IMEX) Runge-Kutta methods for non-hydrostatic atmospheric models

David J. Gardner[1], Jorge E. Guerra[2], François P. Hamon[3], Daniel R. Reynolds[4], Paul A. Ullrich[2], and Carol S. Woodward[1]

[1]Center for Applied Scientific Computing, Lawrence Livermore National Laboratory, 7000 East Ave., Livermore, CA 94550, USA
[2]Department of Land, Air and Water Resources, University of California, Davis, One Shields Ave., Davis, CA 95616, USA
[3]Center for Computational Sciences and Engineering, Lawrence Berkeley National Laboratory, 1 Cyclotron Rd., Berkeley, CA 94720
[4]Department of Mathematics, Southern Methodist University, PO Box 750156, Dallas, TX 75257, USA

*Correspondence to:* David J. Gardner (gardner48@llnl.gov)

**Abstract.** The efficient simulation of non-hydrostatic atmospheric dynamics requires time integration methods capable of overcoming the explicit stability constraints on time step size arising from acoustic waves. In this work we investigate various implicit-explicit (IMEX) additive Runge-Kutta (ARK) methods for evolving acoustic waves implicitly to enable larger time step sizes in a global non-hydrostatic atmospheric model. The IMEX formulations considered include horizontally implicit-vertically implicit (HEVI) approaches as well as splittings that treat some horizontal dynamics implicitly. In each case the impact of solving nonlinear systems in each implicit ARK stage in a linearly implicit fashion is also explored.

The accuracy and efficiency of the IMEX splittings, ARK methods, and solver options are evaluated on a gravity wave and baroclinic wave test case. HEVI splittings that treat some vertical dynamics explicitly do not show a benefit in solution quality or run time over the most implicit HEVI formulation. While splittings that implicitly evolve some horizontal dynamics increase the maximum stable step size of a method, the gains are insufficient to overcome the additional cost of solving a globally coupled system. Solving implicit stage systems in a linearly implicit manner limits the solver cost but this is offset by a reduction in step size to achieve the desired accuracy. Overall, the third order ARS343 and ARK324 methods performed the best, followed by the second order ARS232 and ARK232 methods.

## 1 Introduction

Present-day global climate simulations typically use an atmospheric model configured with a horizontal resolution greater than 10 km. At these scales the equations governing atmospheric motion can utilize the hydrostatic approximation, which assumes a balance between the gravitational and vertical pressure gradient forces and neglects terms related to vertical acceleration and transport of vertical momentum. As a consequence of this simplification, vertically propagating sound waves, which are of little significance in climate studies, are eliminated from the model. This practice is advantageous for computational efficiency with fully explicit time stepping methods, as vertical sound waves impose a stricter stability limit on step size than horizontal sound waves due to the high horizontal to vertical aspect ratio of the mesh. With the most significant constraint on





step size removed, explicit approaches are an attractive option despite their step size limitations from horizontal sound waves. Explicit approaches are employed because of their ease of implementation, the locality of computations, and minimal parallel communication. However, in the near future, increased computational power will enable global climate simulations at scales beyond the hydrostatic limit where vertical acceleration cannot be ignored. At these high resolutions, new model formulations

and numerical methods are needed in order to overcome the computational limitations arising from the fastest waves in the atmosphere.

Accurately modeling atmospheric phenomena at horizontal resolutions below 10 km necessitates moving to a non-hydrostatic formulation of the governing equations. The step size constraints from sound waves can be addressed either by removing the fast waves in the model with a soundproof formulation of the equations or using a numerical method that can stably step

over the fastest waves. The latter approach includes split-explicit (e.g., Klemp et al., 2007), implicit-explicit (IMEX) (e.g., Ullrich and Jablonowski, 2012), and fully implicit (e.g., Yang and Cai, 2014; Yang et al., 2016) time stepping methods. Fully implicit methods enable time steps sizes dictated by the time scales of the processes of interest rather than the stability of the fastest propagating waves. However, achieving good scalability with these methods can be quite challenging as they require optimized nonlinear solvers and preconditioners to efficiently compute the solution of globally coupled nonlinear systems. As

such, split-explicit and IMEX approaches present a potentially simpler alternative as only a subset of the dynamics is treated implicitly. These approaches allow for specialized solvers that can exploit properties of the implicit system at the cost of being stability-limited by the fastest waves in the explicit portion of the splitting.

Split-explicit methods typically divide the dynamics into three groups, fast vertical waves that are treated implicitly, fast horizontal waves that are sub-stepped relative to the other dynamics with an explicit method, and slow dynamics that are updated

with an explicit method using a long time step (e.g., Klemp et al., 2007). Similarly, IMEX methods partition the dynamics into two parts, nonstiff terms that are explicitly updated and stiff terms that are implicitly solved. In a horizontally explicit vertically implicit (HEVI) IMEX approach all horizontal motion is treated explicitly and vertical dynamics are updated implicitly. By solving only the vertical dynamics implicitly, both split-explicit and HEVI methods take advantage of the two-dimensional horizontal domain decomposition of atmospheric models to avoid communication between parallel processes. Since each pro-

cess owns a subset of the vertical columns in the global domain no message passing is necessary during the vertically implicit solves. Split-explicit approaches are able to gain some additional efficiency by sub-stepping the fast horizontal terms while step sizes with HEVI methods are limited by the fastest horizontal dynamics. This restriction can be overcome by incorporating some horizontal terms into the implicit partition at the cost of solving a larger, globally coupled system requiring interprocessor communication during the implicit solve. If the increase in stable step size is sufficiently large, these methods may be able to

overcome the additional expense from parallel communication with an efficient nonlinear solver.

With the push toward exascale computing, there has been increasing interest in evaluating the potential of IMEX methods for efficiently simulating atmospheric dynamics at high resolution. Ullrich and Jablonowski (2012) presented results using a Runge-Kutta-Rosenbrock and Strang-carryover IMEX approach for integrating a non-hydrostatic model in Cartesian geometry. A new second order Runge-Kutta IMEX method is presented in Giraldo et al. (2013) and compared with Runge-Kutta

and multistep IMEX integration schemes from the literature in the non-hydrostatic unified model of the atmosphere (NUMA).



Using NUMA, the accuracy and efficiency of the integration methods were evaluated with one- and three-dimensional linear IMEX splittings by simulating a two-dimensional rising thermal bubble and inertia-gravity waves in three-dimensional Cartesian and spherical domains. Weller et al. (2013) examined the stability properties of twelve Runge-Kutta IMEX methods and compared the accuracy of the methods with two HEVI splittings against a semi-implicit approach using the two-dimensional

compressible Boussinesq equations. The work of Lock et al. (2014) performs a detailed analysis of the same IMEX methods from Weller et al. (2013) on linear scalar and two-dimensional wave equations.

In this work we investigate the performance of nineteen Runge-Kutta IMEX methods from the literature, including many of those tested in Ullrich and Jablonowski (2012), Giraldo et al. (2013), Weller et al. (2013), and Lock et al. (2014), on a nonhydrostatic atmospheric dynamical core using different implicit-explicit splittings of the governing equations and approaches

for solving the nonlinear systems. Methods tested in Weller et al. (2013) and Lock et al. (2014) are evaluated with a three-dimensional fully-compressible set of governing equations that differs from those considered in Giraldo et al. (2013) in terms of formulation, discretization, and approach to implicit-explicit splitting. The Runge-Kutta methods tested in Giraldo et al. (2013) are also included in this study along with additional methods from the literature not considered in the previously cited works. The linearly implicit Runge-Kutta-Rosenbrock approach utilized in Ullrich and Jablonowski (2012) is also compared

against a Newton iteration for solving the nonlinear systems that arise in each implicit Runge-Kutta stage.

The choices of IMEX partitioning, integration method, and implicit solver are evaluated in terms of accuracy and efficiency using the Tempest non-hydrostatic dynamical core (Ullrich, 2014) to determine the optimal combination. Tempest is a flexible global modeling framework for evaluating numerical methods for next-generation, high resolution climate simulations on high-performance computing systems. To ease the exploration of a wide variety of splitting choices and integration schemes

we have interfaced Tempest with the ARKode package of Additive Runge-Kutta (ARK) methods (Reynolds et al.) from the SUNDIALS library (Hindmarsh et al., 2005; SUNDIALS: SUite of Nonlinear DIfferential/ALgebraic equation Solvers web page) of algebraic and differential equations solvers. ARKode is an adaptive-step time integration package for solving initial value problems with fully explicit, fully implicit, or IMEX Runge-Kutta methods. The software framework was designed to be easily incorporated into existing applications and allows significant freedom over the choice of methods and implicit solvers.

In the following section we present the formulation of non-hydrostatic equations implemented in Tempest, followed by a discussion of the spatial and temporal discretizations and splitting approaches in Section 3. The nonlinear and linear solver strategies used with the IMEX methods are covered in Section 4. Numerical experiments and the corresponding results are given in Section 5. A summary of the numerical results, concluding remarks, and directions for future work are given in Section 6.

## 2   Nonhydrostatic Equations

The non-hydrostatic dry-atmosphere shallow-atmosphere equations in the Tempest $\theta$-formulation are in terms of covariant horizontal velocities $u_\alpha$ and $u_\beta$, covariant vertical velocity $u_\xi$, potential temperature $\theta$, and density $\rho$ in an arbitrary coordinate





system $(\alpha, \beta, \xi)$,

$$\frac{\partial u_\alpha}{\partial t} = -\frac{\partial}{\partial \alpha}\left(K + \Phi\right) - \theta \frac{\partial \Pi}{\partial \alpha} + Ju^\beta (fr_\xi^{-1} + \zeta^\xi) - Ju^\xi \zeta^\beta, \tag{1}$$

$$\frac{\partial u_\beta}{\partial t} = -\frac{\partial}{\partial \beta}\left(K + \Phi\right) - \theta \frac{\partial \Pi}{\partial \beta} - Ju^\alpha (fr_\xi^{-1} + \zeta^\xi) + Ju^\xi \zeta^\alpha, \tag{2}$$

$$\frac{\partial u_\xi}{\partial t} = -\frac{\partial}{\partial \xi}\left(K + \Phi\right) - \theta \frac{\partial \Pi}{\partial \xi} + J\left(u^\alpha \zeta^\beta - u^\beta \zeta^\alpha\right), \tag{3}$$

$$\frac{\partial \theta}{\partial t} = -u^\alpha \frac{\partial \theta}{\partial \alpha} - u^\beta \frac{\partial \theta}{\partial \beta} - u^\xi \frac{\partial \theta}{\partial \xi}, \tag{4}$$

$$\frac{\partial \rho}{\partial t} = -\frac{1}{J}\frac{\partial}{\partial \alpha}\left(J\rho u^\alpha\right) - \frac{1}{J}\frac{\partial}{\partial \beta}\left(J\rho u^\beta\right) - \frac{1}{J}\frac{\partial}{\partial \xi}\left(J\rho u^\xi\right). \tag{5}$$

We refer to the system defined by (1) – (5) as the $\theta$–formulation of the non-hydrostatic equations since the thermodynamic equation (4) is expressed in terms of the potential temperature. The conversion between contravariant and covariant velocity components is given as

$$u_i = g_{i\alpha}u^\alpha + g_{i\beta}u^\beta + g_{i\xi}u^\xi \tag{6}$$

$$u^i = g^{i\alpha}u_\alpha + g^{i\beta}u_\beta + g^{i\xi}u_\xi. \tag{7}$$

where $g_{ij}$ and $g^{ij}$ are the covariant and contravariant metric tensors specified in terrain following Cartesian or Spherical coordinates and $J$ is the metric Jacobian defined as

$$J = \sqrt{\det g_{ij}}. \tag{8}$$

$\Phi$ is the product of the gravity constant and elevation $r$, $r_\xi = (\partial r / \partial \xi)$ is the vertical coordinate transform, and $f = 2\Omega \sin \varphi$ is the Coriolis parameter. $K$ is the specific kinetic energy, defined as

$$K = \frac{1}{2}\left(u_\alpha u^\alpha + u_\beta u^\beta + u_\xi u^\xi\right).$$

and $\Pi$ is the Exner pressure function defined as

$$\Pi = c_p \left(\frac{p_0}{p}\right)^{R_d/c_p} = c_p \left(\frac{R_d \rho \theta}{p_0}\right)^{R_d/c_v},$$

where $R_d$, $c_p$, and $p_0$ are the gas constant for dry air, specific heat at constant pressure, and the reference pressure (here chosen to be $10^5$ Pa). The relative vorticity vector is

$$\zeta = \frac{1}{J}\left[\left(\frac{\partial u_\xi}{\partial \beta} - \frac{\partial u_\beta}{\partial \xi}\right)\mathbf{g}_\alpha + \left(\frac{\partial u_\alpha}{\partial \xi} - \frac{\partial u_\xi}{\partial \alpha}\right)\mathbf{g}_\beta + \left(\frac{\partial u_\beta}{\partial \alpha} - \frac{\partial u_\alpha}{\partial \beta}\right)\mathbf{g}_\xi\right]. \tag{9}$$



## 3 Discretization

The non-hydrostatic equations are discretized using a method of lines approach. First, the terms on the right hand sides of (1) – (5) are discretized in space, and then the resulting system of coupled ordinary differential equations is advanced in time with a numerical integration scheme. This two-step process is detailed in the following sections.

### 3.1 Spatial Discretization

The spatial discretization of equations (1) – (5) follows Guerra and Ullrich (2016) where a $4^{th}$ order spectral element method is used for horizontal derivatives in $\alpha$ and $\beta$, and the staggered finite element method is used in $\xi$. Unless otherwise stated, test cases in this work will be configured with Lorenz vertical staggering (vertical velocity computed and stored at interfaces including model boundaries) and regular grid distribution in each column.

In Tempest, hyperviscosity is employed in the horizontal directions by default. The operators are $4^{th}$ order derivatives with nominal coefficients of $1.0 \times 10^{15}$ $m^2 s^{-1}$ following Guerra and Ullrich (2016) and Ullrich (2014). In particular, the use of hyperviscosity corrects dispersive errors and ringing associated with computational modes due to accumulation of energy near the grid truncation scale. The use of hyperviscosity is necessary since the spectral element discretization implicitly conserves energy (Taylor and Fournier, 2010), and hence provides no mechanism for implicit diffusion of energy at short wavelengths. It is important to note that hyperviscosity is applied at the end of each time step as a separate forward Euler update. As such, it is not part of the additive Runge-Kutta method used to integrate the equations. Additionally, vertical upwinding is applied to the horizontal velocities, potential temperature, and density.

### 3.2 Temporal Discretization

There are numerous approaches for integrating the system of ODEs resulting from the spatial discretization of (1) – (5), including multistep or multistage methods that treat the system in a fully explicit, fully implicit, or split implicit-explicit manner. This work focuses on the application of multistage IMEX integrators defined by additive Runge-Kutta (ARK) methods that split the right hand side into two parts, an explicit (nonstiff) and an implicit (stiff) part. Section 3.2.1 presents the general formulation of ARK methods. The various options explored for partitioning terms in the non-hydrostatic equations into implicit and explicit parts are presented in Sections 3.2.2 and 3.2.3.

### 3.2.1 Additive Runge-Kutta Methods

The spatially discretized non-hydrostatic equations can be written as a general initial value problem with the right hand side additively split into two parts,

$$\frac{d\boldsymbol{y}}{dt} = \boldsymbol{f}^E(t, \boldsymbol{y}) + \boldsymbol{f}^I(t, \boldsymbol{y}), \qquad \boldsymbol{y}(t_0) = \boldsymbol{y}_0. \tag{10}$$





The model state vector is $\boldsymbol{y} = (u_\alpha, u_\beta, w, \theta, \rho)^T$ in the $\theta$-formulation. Under this notation, $\boldsymbol{f}^E$ and $\boldsymbol{f}^I$ correspond to the spatial terms that will be integrated explicitly and implicitly respectively, and $\boldsymbol{y}_0$ is the initial state at time $t_0$. The system (10) is evolved from time $t_{n-1}$ to time $t_n$ using ARK methods of the form

$$\boldsymbol{z}_i = \boldsymbol{y}_{n-1} + h_n \sum_{j=1}^{i-1} a_{i,j}^E \boldsymbol{f}^E(t_{n,j}^E, \boldsymbol{z}_j) + h_n \sum_{j=1}^{i} a_{i,j}^I \boldsymbol{f}^I(t_{n,j}^I, \boldsymbol{z}_j), \quad i = 1, \ldots, s, \tag{11}$$

$$\boldsymbol{y}_n = \boldsymbol{y}_{n-1} + h_n \sum_{i=1}^{s} \left( b_i^E \boldsymbol{f}^E(t_{n,i}^E, \boldsymbol{z}_i) + b_i^I \boldsymbol{f}^I(t_{n,i}^I, \boldsymbol{z}_i) \right), \tag{12}$$

where $\boldsymbol{y}_n$ is an approximation of $\boldsymbol{y}(t_n)$, $\boldsymbol{z}_i$ is an intermediate stage solution in an ARK method with $s$ stages, $h_n = t_n - t_{n-1}$ is the time step size, and $t_{n,i}^E = t_{n-1} + c_i^E h_n$ and $t_{n,i}^I = t_{n-1} + c_i^I h_n$ are intermediate stage times. Several of the methods we examine include an embedded solution,

$$\tilde{\boldsymbol{y}}_n = \boldsymbol{y}_{n-1} + h_n \sum_{i=1}^{s} \left( \tilde{b}_i^E \boldsymbol{f}^E(t_{n,i}^E, \boldsymbol{z}_i) + \tilde{b}_i^I \boldsymbol{f}^I(t_{n,i}^I, \boldsymbol{z}_i) \right), \tag{13}$$

for estimating the local truncation error to adapt the time step size. The numerical studies that follow use a fixed time step size and thus do not utilize a local error estimate. However, methods with embeddings are of particular interest as we will explore leveraging variable time step sizes in subsequent work.

A particular ARK method is defined by a combination of an explicit and a diagonally implicit pair of Butcher tableaux,

$$
\begin{array}{c|c}
\boldsymbol{c}^E & A^E \\ \hline
\boldsymbol{b}^E \\
\tilde{\boldsymbol{b}}^E
\end{array}
=
\begin{array}{c|cccccc}
& 0 & 0 & 0 & \cdots & 0 \\
c_2^E & a_{2,1}^E & 0 & 0 & \cdots & 0 \\
c_3^E & a_{3,1}^E & a_{3,2}^E & 0 & \cdots & 0 \\
\vdots & \vdots & & & \ddots & 0 \\
c_s^E & a_{s,1}^E & a_{s,2}^E & \cdots & a_{s,s-1}^E & 0 \\ \hline
& b_1^E & b_2^E & \cdots & b_{s-1}^E & b_s^E \\
& \tilde{b}_1^E & \tilde{b}_2^E & \cdots & \tilde{b}_{s-1}^E & \tilde{b}_s^E
\end{array},
\qquad
\begin{array}{c|c}
\boldsymbol{c}^I & A^I \\ \hline
\boldsymbol{b}^I \\
\tilde{\boldsymbol{b}}^I
\end{array}
=
\begin{array}{c|ccccc}
c_1^I & a_{1,1}^I & 0 & 0 & \cdots & 0 \\
c_2^I & a_{2,1}^I & a_{2,2}^I & 0 & \cdots & 0 \\
c_3^I & a_{3,1}^I & a_{3,2}^I & a_{3,3}^I & \cdots & 0 \\
\vdots & \vdots & & & \ddots & 0 \\
c_s^I & a_{s,1}^I & a_{s,2}^I & \cdots & a_{s,s-1}^I & a_{s,s}^I \\ \hline
& b_1^I & b_2^I & \cdots & b_{s-1}^I & b_s^I \\
& \tilde{b}_1^I & \tilde{b}_2^I & \cdots & \tilde{b}_{s-1}^I & \tilde{b}_s^I
\end{array}.
\tag{14}
$$

When $a_{i,i}^I \neq 0$, computing the stage value $\boldsymbol{z}_i$ requires solving a (non)linear system of equations. This system, and approaches for computing the stage solutions, are discussed in Section 4.

While (11) – (13) define an ARK method in terms of a linear combination of right hand side evaluations at internal stage values, we note that it is customary in the climate modeling community to cast ARK schemes as linear combinations of states produced from explicit and implicit Euler steps. There, the objective is to store state vectors only and make single explicit/implicit function evaluations at a given stage. Since the stage coefficient matrices in (14) are lower triangular (strictly so for the explicit coefficients), then any preceding function evaluations may be written in terms of preceding state vectors and substituted into a current stage. While the two approaches are entirely equivalent, we present all ARK methods and their



corresponding tables in standard form (11) – (14), both to connect with the literature defining each method, and since we use this form in computing our results.

We investigate a number of ARK methods from the literature with a variety of numerical properties:

- Classical second (ARS232, ARS222, and ARS233) and third (ARS343 and ARS443) order methods from Ascher et al. (1997).

- The third (ARK324), fourth (ARK436), and fifth (ARK548) order methods from Kennedy and Carpenter (2003).

- The second order ARK232 method derived for the NUMA model and presented in Giraldo et al. (2013).

- Second (SSP2(222), SSP2(332)a, and SSP3(332)) and third (SSP3(433)) order strong stability preserving methods from Pareschi and Russo (2005).

- Second order strong stability preserving method SSP2(332)b from Higueras (2006), the optimized second order methods SSP2(332)lpm1, SSP2(332)lpm2, SSP2(332)lpum, SSP2(332)lspum from Higueras et al. (2014), and the third order method SSP3(333) from Higueras (2009).

The ARK232, ARS232, ARS233, ARS443, SSP2(222), SSP2(332)a, SSP3(332) and SSP3(433) methods were previously examined by Weller et al. (2013) with different splittings on two vertical slice cases of the compressible Boussinesq equations. In these tests the ARK232 method presented in Giraldo et al. (2013) had the best overall performance. Giraldo et al. (2013) also compared ARK232 with ARK324 and ARK436 with different splittings on a 2D rising thermal bubble test in Cartesian coordinates and 3D inertia-gravity wave tests in Cartesian coordinates and on the sphere. The ARK324 and ARK436 methods were most efficient when high accuracy is required and ARK232 had greater efficiency when less accuracy was required. In addition to the method tested in Weller et al. (2013) and Giraldo et al. (2013) we include the ARS222, ARS343, ARK548, and SSP methods from Higueras (2006, 2009) and Higueras et al. (2014).

All of the above methods, with the exception of ARS233 and SSP3(333), are constructed with an L-stable implicit method. Thus, the implicit portion of the method is accurate in the limit of the stiff term becoming infinitely fast, meaning that slow dynamics are resolved while fast modes, e.g. acoustic waves, are damped. Two methods, ARS233 and SSP2(222), are B-stable which is a nonlinear stability indicating that the difference between two numerical solutions does not increase with time. Several methods are strong stability preserving (SSP) and are designed to maintain the total variation diminishing (TVD) property of a spatial discretization. The optimized SSP methods from Higueras et al. (2014) consider additional properties beyond optimizing the region of absolute monotonicity in SSP schemes. Ten of the methods considered are stiffly accurate, that is $a_{s,i}^I = b_i^I$, and two of those methods, ARS222 and ARS443, also have $a_{s,i}^E = b_i^E$. Ascher et al. (1997) notes that having the both $a_{s,i}^I = b_i^I$ and $a_{s,i}^E = b_i^E$ so that $\boldsymbol{z}_s = \boldsymbol{y}_n$ is useful for very stiff problems. However, it is unclear why this property is beneficial and the two methods with this property do not outperform other methods in the two test cases considered. All methods, except ARS222 and ARS443, have the same $b_i$ values for the explicit and implicit methods so that the $\boldsymbol{f}^E$ and $\boldsymbol{f}^I$ functions are weighted equally at the same stage solution. Only the SSP methods have different explicit and implicit stage





times (i.e., $c^E \neq c^I$). Finally, the methods of Kennedy and Carpenter (2003) and Giraldo et al. (2013) have implicit methods with second order stage accuracy to limit order reduction when applied to stiff systems. Appendix A contains a summary of different properties of the ARK methods considered in this work.

### 3.2.2 Horizontally explicit vertically implicit splittings

In this section we present four horizontally explicit vertically implicit (HEVI) formulations of the non-hydrostatic equations (1) – (5) in which the horizontal terms are evaluated explicitly and some of the vertical terms are solved implicitly. The partitioning of terms into the explicit or implicit right hand sides are given by

$$\frac{\partial u_\alpha}{\partial t} = -\frac{\partial}{\partial \alpha}\left(K + \Phi\right) - \theta\frac{\partial \Pi}{\partial \alpha} + Ju^\beta(fr_\xi^{-1} + \zeta^\xi) - Ju^\xi\zeta^\beta, \tag{15}$$

$$\frac{\partial u_\beta}{\partial t} = -\frac{\partial}{\partial \beta}\left(K + \Phi\right) - \theta\frac{\partial \Pi}{\partial \beta} - Ju^\alpha(fr_\xi^{-1} + \zeta^\xi) + Ju^\xi\zeta^\alpha, \tag{16}$$

$$\frac{\partial u_\xi}{\partial t} = \underbrace{-\frac{\partial K}{\partial \xi} + u^\alpha\frac{\partial u_\alpha}{\partial \xi} + u^\beta\frac{\partial u_\beta}{\partial \xi}}_{\text{imp (A, C) / exp (B, D)}} \underbrace{-\frac{\partial \Phi}{\partial \xi} - \theta\frac{\partial \Pi}{\partial \xi}}_{\text{imp (A, B, C, D)}} - u^\alpha\frac{\partial u_\xi}{\partial \alpha} - u^\beta\frac{\partial u_\xi}{\partial \beta}, \tag{17}$$

$$\frac{\partial \theta}{\partial t} = -u^\alpha\frac{\partial \theta}{\partial \alpha} - u^\beta\frac{\partial \theta}{\partial \beta} \underbrace{-u^\xi\frac{\partial \theta}{\partial \xi}}_{\text{imp (A, B) / exp (C, D)}}, \tag{18}$$

$$\frac{\partial \rho}{\partial t} = -\frac{1}{J}\frac{\partial}{\partial \alpha}\left(J\rho u^\alpha\right) - \frac{1}{J}\frac{\partial}{\partial \beta}\left(J\rho u^\beta\right)\underbrace{-\frac{1}{J}\frac{\partial}{\partial \xi}\left(J\rho u^\xi\right)}_{\text{imp (A, B, C, D)}}. \tag{19}$$

The choice of an explicit or implicit treatment of these terms is guided by two core requirements: First, we require that the terms responsible for vertically propagating sound waves (namely, the buoyancy term, $-\frac{\partial \Phi}{\partial \xi} - \theta\frac{\partial \Pi}{\partial \xi}$, in the vertical velocity

equation (17) and the vertical flux term, $-\frac{1}{J}\frac{\partial}{\partial \xi}\left(J\rho u^\xi\right)$, in the density equation (19)) be handled implicitly. Treating these terms explicitly would leave us bound to the CFL condition for vertically propagating sound waves (around 2s for the simulations in this paper) and so would not lead to a computationally competitive scheme. Second, all terms associated with vertical momentum transport (the first three terms in (17)) must be handled together. In practice, these terms cancel each other nearly exactly and so splitting them can dramatically impact model stability. These terms can be handled either implicitly or explicitly,

as they are associated with the vertical advective speed, and hence an explicit treatment does not impact model stability. Whereas an explicit treatment of these terms is generally simpler, in combination with the last term in (19) they are together responsible for vertical Kinetic energy transport. Consequently, it could be argued that these terms should be handled using the same discretization as the vertical mass transport term in order to ensure energy conservation by the vertically implicit update. In this study, the impact of an explicit and implicit treatment of the vertical transport of $\theta$ is also explored (the last term in

(18)). Although sound waves are not expressed in the $\theta$ field, there is a substantial difference in model stability that emerges from whether this term is treated implicitly or explicitly.



Based on the principles above, terms in (15) – (19) without underbraces are always evaluated explicitly and those with underbraces are treated either implicitly or explicitly depending on the particular HEVI formulation. In order from most implicit to least implicit, we consider:

1. HEVI–A, all vertical dynamics except vertical advection of horizontal velocity in (15) and (16) are treated implicitly,

2. HEVI–B, vertical velocity advection in (17) is treated explicitly,

3. HEVI–C, thermodynamic advection in (18) is treated explicitly, and

4. HEVI–D, vertical velocity advection in (17) and thermodynamic advection in (18) are treated explicitly.

### 3.2.3 IMEX splittings with horizontally implicit terms

In addition to the HEVI options, we also consider IMEX splittings that solve various parts of horizontal dynamics implicitly.
These formulations contain the same vertically implicit terms as HEVI–A but add some of the horizontal terms into the implicit function,

$$\frac{\partial u_\alpha}{\partial t} = -\frac{\partial}{\partial \alpha}(K+\Phi) \underbrace{-\theta\frac{\partial \Pi}{\partial \alpha}}_{\text{imp(B) / exp(A)}} + Ju^\beta(fr_\xi^{-1}+\zeta^\xi) - Ju^\xi\zeta^\beta, \tag{20}$$

$$\frac{\partial u_\beta}{\partial t} = -\frac{\partial}{\partial \beta}(K+\Phi) \underbrace{-\theta\frac{\partial \Pi}{\partial \beta}}_{\text{imp(B) / exp(A)}} - Ju^\alpha(fr_\xi^{-1}+\zeta^\xi) + Ju^\xi\zeta^\alpha, \tag{21}$$

$$\frac{\partial u_\xi}{\partial t} = \underbrace{-\frac{\partial}{\partial \xi}(K+\Phi) -\theta\frac{\partial \Pi}{\partial \xi} + u^\alpha\frac{\partial u_\alpha}{\partial \xi} + u^\beta\frac{\partial u_\beta}{\partial \xi}}_{\text{imp(A, B)}} - u^\alpha\frac{\partial u_\xi}{\partial \alpha} - u^\beta\frac{\partial u_\xi}{\partial \beta}, \tag{22}$$

$$\frac{\partial \theta}{\partial t} = \underbrace{-u^\alpha\frac{\partial \theta}{\partial \alpha} - u^\beta\frac{\partial \theta}{\partial \beta}}_{\text{imp(B) / exp(A)}} \underbrace{-u^\xi\frac{\partial \theta}{\partial \xi}}_{\text{imp}}, \tag{23}$$

$$\frac{\partial \rho}{\partial t} = \underbrace{-\frac{1}{J}\frac{\partial}{\partial \alpha}(J\rho u^\alpha) - \frac{1}{J}\frac{\partial}{\partial \beta}(J\rho u^\beta) - \frac{1}{J}\frac{\partial}{\partial \xi}(J\rho u^\xi)}_{\text{imp(A, B)}} \tag{24}$$

As before, terms without underbraces in (20) – (24) are treated explicitly. We examine two configurations with horizontally implicit terms:

1. IMEX–A, the density equation (24) is fully implicit, and

2. IMEX–B, the density (24), thermodynamics (23), and Exner pressure in (20) and (21) are solved implicitly.

The IMEX–A option treats the density equation with a single consistent scheme while IMEX–B is motivated by semi-implicit splittings (e.g. Weller et al., 2013) and treats the pressure gradient fully implicitly. By incorporating some of the fast


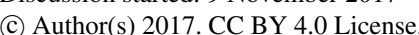


horizontal dynamics into the implicit portion of the splitting, these IMEX formulations may enable larger stable step sizes than are possible with the HEVI options. However, treating horizontal dynamics implicitly also introduces coupling between vertical columns in the implicit solves, and this increased coupling in turn increases the linear solution expense. In the numerical experiments below we will test if the increased steps sizes are enough to offset the additional solver cost.

## 4 Solvers

An $s$-stage ARK method defined by (11) – (13) requires the solution of at most $s$ nonlinear systems of the form

$$\boldsymbol{G}(\boldsymbol{z}_i) \equiv \boldsymbol{z}_i - h_n a^I_{i,i} \boldsymbol{f}^I(t^I_{n,i}, \boldsymbol{z}_i) - \boldsymbol{d}_i = 0, \quad i = 1, \ldots, s, \tag{25}$$

to compute the stage solutions, $\boldsymbol{z}_i$, where

$$\boldsymbol{d}_i \equiv \boldsymbol{y}_{n-1} + h_n \sum_{j=1}^{i-1} \left[ a^E_{i,j} \boldsymbol{f}^E(t^E_{n,j}, z_j) + a^I_{i,j} \boldsymbol{f}^I(t^I_{n,j}, z_j) \right] \tag{26}$$

is known data from previous stage values. The structure of (25) is highly dependent on the underlying splitting, which determines the size of the system and the spatial coupling between the algebraic equations in this system. Therefore, efficient solution strategies that take full advantage of the structure of the nonlinear system resulting from the splitting are highly desirable. This topic is addressed in the following subsections where we present the solver approaches considered in this work.

### 4.1 Nonlinear Solvers

Newton's method finds the solution of (25) using an iterative approach

$$\boldsymbol{z}_i^{(m+1)} = \boldsymbol{z}_i^{(m)} + \boldsymbol{\delta}^{(m+1)} \tag{27}$$

where $m \geq 0$ is the iteration index and the update $\boldsymbol{\delta}^{(m+1)}$ is the solution of the linear system

$$\mathcal{A}(\boldsymbol{z}_i^{(m)}) \boldsymbol{\delta}^{(m+1)} = -\boldsymbol{G}(z_i^{(m)}) \tag{28}$$

obtained from a linearization of (25),

$$\mathcal{A}(\boldsymbol{z}_i^{(m)}) \equiv I - h_n a^I_{i,i} \mathcal{J}(\boldsymbol{z}_i^{(m)}) \tag{29}$$





in which $\mathcal{J}$ is the Jacobian matrix of $\boldsymbol{f}^I$ evaluated at the current iteration. Following the ODE literature, we consider the iteration converged when

$$R_i \|\boldsymbol{\delta}^{(m+1)}\| < \epsilon \tag{30}$$

where $R_i$ is an estimate of the linear convergence rate, $\|\cdot\|$ is a weighted root-mean-square (WRMS) norm, and $\epsilon$ is the
nonlinear tolerance (Hindmarsh et al., 2005). The convergence rate estimate $R_i$ is initialized to 1 and for $m > 0$ is updated as

$$R_i = \max\left(0.3 R_i, \frac{\|\boldsymbol{\delta}^{(m+1)}\|}{\|\boldsymbol{\delta}^{(m)}\|}\right). \tag{31}$$

For a vector $\boldsymbol{v}$ with length $N$, the WRMS is norm defined as

$$\|\boldsymbol{v}\| = \left(\frac{1}{N}\sum_{i=1}^{N}(w_i v_i)^2\right)^{1/2}, \qquad \text{with} \qquad w_i = \frac{1}{\epsilon_r |v_i| + \epsilon_a}, \tag{32}$$

where $\epsilon_a$ and $\epsilon_r$ are the absolute and relative tolerances for the time evolved solution, respectively. With this choice of weight-
ing, a WRMS norm of 1 is considered small for any error-like quantities since $1/w_i$ represents a tolerance on the components of the solution vector. To keep error in the nonlinear solve from interfering with the time integration error, we use the ARKode default nonlinear tolerance $\epsilon = 10^{-1}$ in equation (30).

Newton's method can be quite expensive, especially when many iterations are needed to achieve convergence, since each iteration involves computing or approximating the Jacobian matrix and performing a linear solve. As an alternative we also
consider treating (25) as a linearly implicit system. This Rosenbrock-like approach, used in Ullrich and Jablonowski (2012) and Guerra and Ullrich (2016), consists of performing a single iteration of Newton's method, thus limiting the cost of the nonlinear solver. However, this approach may produce a lower quality solution when one Newton iteration does not sufficiently solve the original nonlinear problem.

In both solver approaches, the solution value $\boldsymbol{y}_{n-1}$ is utilized as the initial iterate, $\boldsymbol{z}_i^0$, (i.e. the trivial predictor) in the
nonlinear solves. While alternative predictor methods are not explored in this work their impact on the speed and robustness of the nonlinear solve is a topic of future investigation.

### 4.2   Linear Solvers

Finding the solution of the nonlinear system (25) using one or several Newton iterations requires solving the linear system (28) for the iteration update. Since the HEVI splittings treat all the horizontal terms explicitly, (28) does not contain any coupling
between the degrees of freedom in different vertical columns of the atmosphere. That is, the coupling introduced by the implicit terms only acts in the vertical direction, and the linear solve is, therefore, decomposable into a series of independent columnwise solves. The linear solves in each column are performed with the direct banded solver `dgbsv` from the Linear Algebra PACKage (LAPACK) (Anderson et al., 1999) without any need for interprocessor communication. Moreover, when





combined with the Rosenbrock-like approach no communication is necessary in the nonlinear solve, and neither a nonlinear nor a linear tolerance needs to be set.

The inclusion of horizontal dynamics in the implicit function introduces coupling between degrees of freedom located in different columns, and a linear solve over the full domain is necessary to compute the Newton update. In this case we employ a Newton-Krylov approach for the nonlinear solve where an approximate solution of (28) is found using the GMRES method (Saad and Schultz, 1986). Krylov methods require only the action of a matrix on a vector, and this operation is approximated through a finite difference computation

$$
\mathcal{A}\left(z_i^{(m)}\right) v \approx \frac{G\left(z_i^{(m)} + \sigma v\right) - G\left(z_i^{(m)}\right)}{\sigma} \tag{33}
$$

where the increment $\sigma = \|v\|$ to ensure $\|\sigma v\| = 1$. Hence, constructing the full Jacobian matrix is unnecessary. We additionally precondition the GMRES solver on the right, using the HEVI-based columnwise direct solves described above. Since the HEVI methods treat only vertical dynamics implicitly the horizontal dynamics in the IMEX splittings remain unpreconditioned. As GMRES is iterative, we consider the linear solution to be converged when the preconditioned residual vector $r$ satisfies

$$
\|r\| \leq 0.1 \epsilon_L \epsilon \tag{34}
$$

where $\|\cdot\|$ is the WRMS norm. Like with the nonlinear solver error, the error in the linear solve must also be controlled to not interfere with the integration error; we therefore utilize the ARKode default value of $\epsilon_L = 0.05$.

## 5 Numerical results

We evaluate the accuracy and computational efficiency of the various implicit-explicit splittings, ARK methods, and solver options on two test cases. Section 5.1 presents results for the propagation of gravity waves on a sphere on a reduced-radius planet, and Section 5.2 focuses on the development of a baroclinic wave. Simulations are preformed on the Cab Linux computing cluster at Lawrence Livermore National Laboratory. Each Cab node consists of two Intel Xeon 8-core SandyBridge processors with 32 GB of memory per node. All tests are run on 6 compute nodes using 96 MPI tasks. The absolute and relative tolerances in the numerical experiments are $\epsilon_a = \epsilon_r = 10^{-4}$. These tolerances where chosen to produce results with ARKode that matched the solutions obtained with the native Tempest implementation of ARS232. The maximum number of Newton iterations is set to 10, and the maximum number of GMRES iterations is set to 50, although these maximum values were only attained in one combination of splitting and solver as noted below.

### 5.1 Gravity Wave

The gravity wave test as defined in Ullrich et al. (2012) begins with an initially balanced atmosphere on a reduced radius Earth (1/125 in size). A small potential temperature perturbation is added to the initial state causing the development of gravity





waves. The domain is discretized using 2,400 elements and 10 vertical levels. The test is simulated for one hour with time steps of 0.01s, 0.1s, 0.5s, 1s, 2s, 4s, and 8s with each of the different splittings, methods, and solvers described above. To compare the accuracy and efficiency of the different options the RMS error of the state vector with respect to a reference solution is computed at the final time. The reference solution is computed using a step size of 0.001s with a fully explicit third order

five-stage Runge-Kutta method (KGU35) derived by Ullrich and implemented in Tempest (Guerra and Ullrich, 2016). This particular explicit method was created using the stability optimization presented in Kinnmark and Gray (1984) to maximize the stability region along the imaginary axis.

Accuracy and efficiency plots are shown in Figures 1 – 6 for the gravity wave test. With the exception of the IMEX–B splitting, as noted below, the choice between a Rosenbrock-like approach or a full Newton iteration to solve the stage systems

does not impact the maximum stable time step size of a given splitting or method, and both solver approaches produce nearly identical errors for this test case. Thus using only a single Newton iteration provides a sufficiently accurate solution to the nonlinear stage systems in each time step. The Newton solver also consistently increases computational cost by approximately 20% to 50% over the Rosenbrock-like results with HEVI splittings and adds an additional cost of at least 10% with the IMEX options. Since there is not a significant benefit from a Newton solver in this test case, Figures 1 – 6 focus only on results

with a Rosenbrock-like approach. The choice of nonlinear solver is more important in the baroclinic wave test case and will be discussed further in the following section. Additionally, treating the vertical velocity or thermodynamic advection explicitly has a negligible impact on the solution error and integrator efficiency, so results with HEVI–B, C, and D are indistinguishable from those of HEVI–A. As such Figures 1 – 6 present only HEVI–A results and any conclusions on the behavior or performance of HEVI–A also apply to HEVI–B, C, and D.

The second order ARK methods can be divided into two groups based on accuracy regardless of the splitting choice. The more accurate group of methods consists of the lpm1, lpm2, lpum, and lspum optimized variants of SSP2(332) from Higueras et al. (2014), and the remaining second order schemes comprise the second group with slightly less accuracy.

The approximate largest stable step size is consistent across the HEVI splittings. The ARK232, ARS232, and SSP3(332) methods are stable with $h_n = 2$s, the SSP2(332) methods are stable with $h_n = 1$s, and the remaining two methods are stable

with $h_n = 0.5$s. With the IMEX–A option, all of the methods are able to achieve a step size of 2s, and including more implicit terms in IMEX–B increases the maximum step size to 8s for all of the methods with the Rosenbrock-like approach. The IMEX–B splitting is the only case where integrator behavior differs when using the Newton solver rather than the Rosenbrock-like approach. In this instance the maximum stable time step is smaller, approximately 1/4 the step size or smaller depending on the method, with the Newton iteration as it is unable to converge to the given tolerance with larger step sizes. Such behavior may

be due to using the trivial predictor and more sophisticated approaches could improve convergence with the Newton solver. Given the high cost of the IMEX–B splitting with Rosenbrock-like approach compared to the HEVI methods the evaluation of alternative predictors with the Newton solver is left to future work.

The relative efficiency of the different ARK methods is also consistent across the splitting options. Despite requiring three implicit solves per time step, the optimized SSP2(332) methods from Higueras et al. (2014) are the most computationally

efficient second order approaches when higher solution accuracy is desired. Because of the larger maximum stable step size



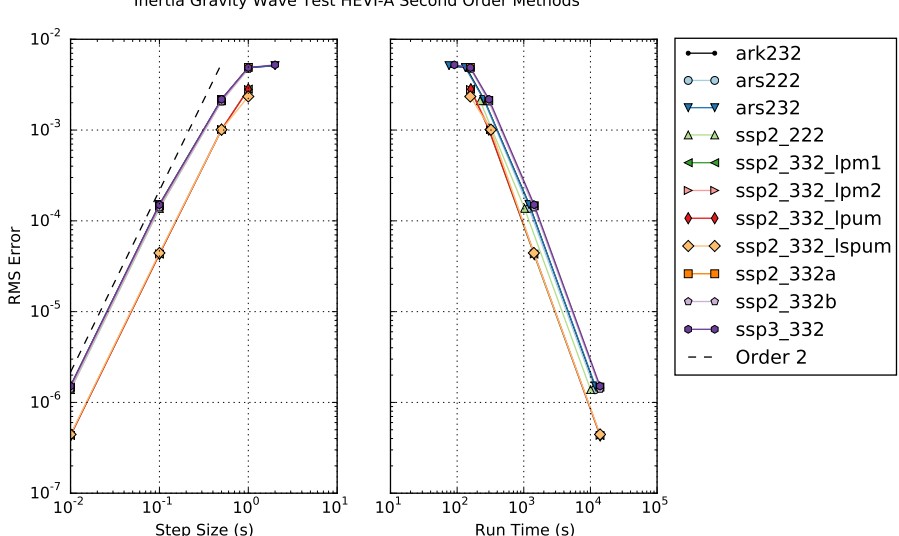

**Figure 1.** Accuracy (left) and efficiency (right) for the second order methods using the Rosenbrock-like approach with the HEVI-A splitting. The dashed line represents second order convergence. The ARK methods fall into two groups with similar accuracy. Results using multiple Newton iterations to compute the stage solutions give nearly identical error values to the Rosenbrock-like results but increase the computational cost by 20% to 50%.

ARK232, ARS232, and SSP3(332) provide slightly faster solution times with the HEVI splittings but with larger error values. Because of the increased stability in the IMEX-A and IMEX-B tests many of the second order methods become competitive with the Higueras et al. (2014) SSP2(332) methods at $h_n = 2$s, but as with the HEVI splittings the optimized SSP2(332) methods are more efficient when greater accuracy is required. Making more terms explicit in the non-hydrostatic equations

does not cause a significant difference in run times between the HEVI options. The inclusion of horizontally implicit terms and the additional communication necessary in each implicit solve with the IMEX–A and IMEX–B options increases the simulation time by approximately 25% – 60% over the HEVI results.

Across the splitting options, the majority of the third order ARK methods produce solutions with approximately the same level of accuracy with the exception of SSP3(433) and ARS233 which are consistently more and less accurate than the other

third order schemes, respectively. The fourth order accurate ARK436 has smaller errors than all second and third order methods and the fifth order ARK548 method generally has the lowest error overall. The fifth order ARK548 does not achieve the expected convergence rate and with the IMEX–A splitting all of the methods drop to second order convergence. Since the IMEX–B and HEVI–A methods show no such deterioration in accuracy, and they match IMEX–A but have more/fewer implicit terms, respectively, we believe that IMEX–A suffers from order reduction in the coupling terms. Specifically, it is likely that

IMEX–A splits two large and opposite terms into explicit and implicit components, whereas IMEX–B and HEVI–A treat both terms consistently. As a result, partial derivatives of $\boldsymbol{f}^E$ and $\boldsymbol{f}^I$ in the IMEX–A splitting may have large magnitudes, resulting in increased stiffness, causing the order reduction.



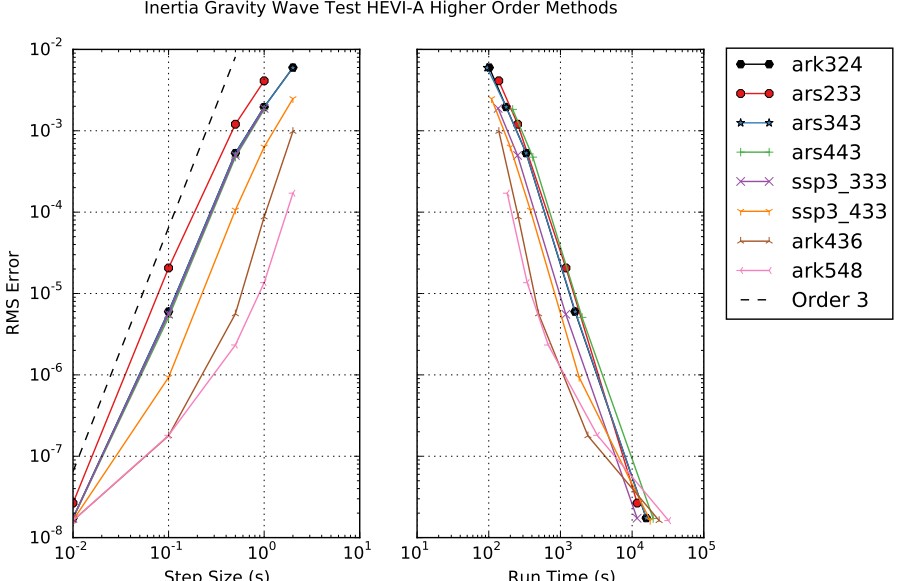

**Figure 2.** Accuracy (left) and efficiency (right) for the third, fourth, and fifth order methods using the Rosenbrock-like approach with the HEVI-A splitting. The dashed line represents third order convergence. Results using multiple Newton iterations to compute the stage solutions give nearly identical error values to the Rosenbrock-like results but increase the computational cost by 20% to 50%.

Like the second order methods, the choice of HEVI splitting does not effect the approximate maximum step size of a given third order method. ARS233, ARS443, and SSP3(333) all have a maximum steps size of 1s and ARS343, SSP3(433), ARK324, ARK436, and ARK548 allow steps of up to 2s. SSP3(333) is the only method to show a doubling in the maximum step size, going from 1s to 2s, due to the additional implicitness in IMEX–A. In the IMEX–B tests all of the methods, with the exception of SSP3(333) which does not gain stability, have an increase in maximum step size to 8s. As with with second order methods the IMEX–B splitting is the only option where the choice of a Rosenbrock-like approach alters the integrator results by reducing the maximum step size due to lack of solver convergence.

Among the third order methods, SSP3(443) is the most efficient method except at the smallest step sizes where convergence begins to slow and SSP3(333) becomes faster for the same accuracy. Likewise the fourth and fifth order methods are more cost effective until the convergence slows at the smallest time step sizes. SSP3(333) is the best approach for lower accuracy levels in IMEX–B and is the best scheme in IMEX–A. For higher accuracy with IMEX–B, the faster convergence of ARK436 and ARK548 make these approaches more efficient until convergence begins to slow at small step sizes. As with the second order methods, HEVI–B, C and D do not present an advantage over HEVI–A in run time, and the additional communication required by the horizontal terms in the implicit portion of the IMEX methods is not offset by sufficient gains in step size.

With both the second and higher order integration methods HEVI–A with the Rosenbrock-like approach is the best combination in this test case. For the most part, third order methods outperform the second order methods in terms of accuracy at a given step size. Since the third order methods do not increase the maximum stable step size over that achieved by the second



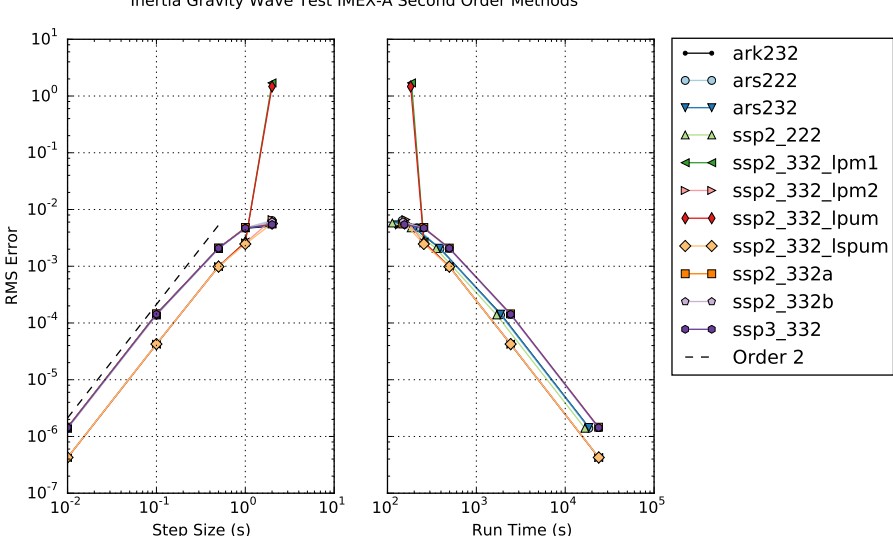

**Figure 3.** Accuracy (left) and efficiency (right) for the second order methods using the Rosenbrock-like approach with the IMEX-A splitting. The ARK methods fall into two groups with similar accuracy. The dashed line represents second order convergence. Results using multiple Newton iterations to compute the stage solutions give nearly identical error values to the Rosenbrock-like results but increase the computational cost by at least 10%.

order methods, the second order schemes are more efficient at looser error requirements and higher order methods are best when more accuracy is necessary.

### 5.2 Baroclinic Wave

The second test case simulates the development of a baroclinic wave over the course of approximately 10 days as described in Ullrich et al. (2014). For this test case, we focus on how the methods, splittings, and solvers perform near the maximum stable time step size in a 30 day simulation. The domain is discretized with 2,400 elements and 30 vertical levels. Starting from a step size of 100s, $h_n$ is increased, using steps that evenly divide one day, until the method is unable to simulate 30 days without a solver failure. Table 1 lists the approximate largest step sizes for each of the methods. As with the results in the gravity wave test, the choice of a Rosenbrock-like or Newton solver does not generally impact the largest stable step size for a given splitting or method with the exception of four methods (ARS222, SSP2(332)lspum, ARS233, and SSP3(433)). However, the solver selected does affect the quality of the solution produced at large time step sizes and in many cases a smaller step size may be necessary to compute a sufficiently accurate solution with the Rosenbrock-like approach.

Since this problem produces a strong instability, comparisons against a highly resolved reference solution, as was used in the gravity wave test, do not yield a good metric for quality of a numerical solution. To define an acceptable numerical solution generated by the methods at any given time step, the results of the implicit-explicit simulations (HEVI or IMEX) are compared against the range of maximum vertical velocities produced by five explicit simulations with initial conditions





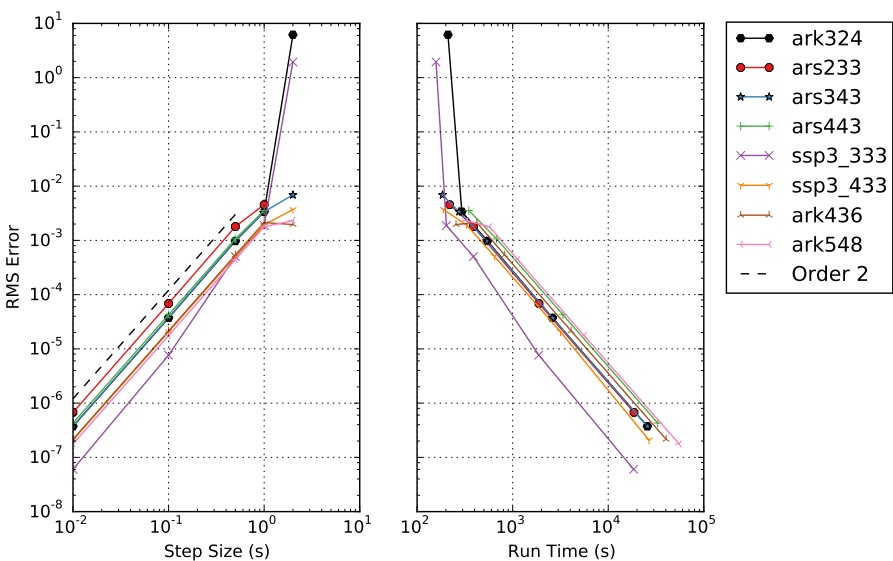

**Figure 4.** Accuracy (left) and efficiency (right) for the third, fourth, and fifth order methods using the Rosenbrock-like approach with the IMEX-A splitting. The dashed line represents second order convergence. Results using multiple Newton iterations to compute the stage solutions give nearly identical error values to the Rosenbrock-like results but increase the computational cost by at least 10%.

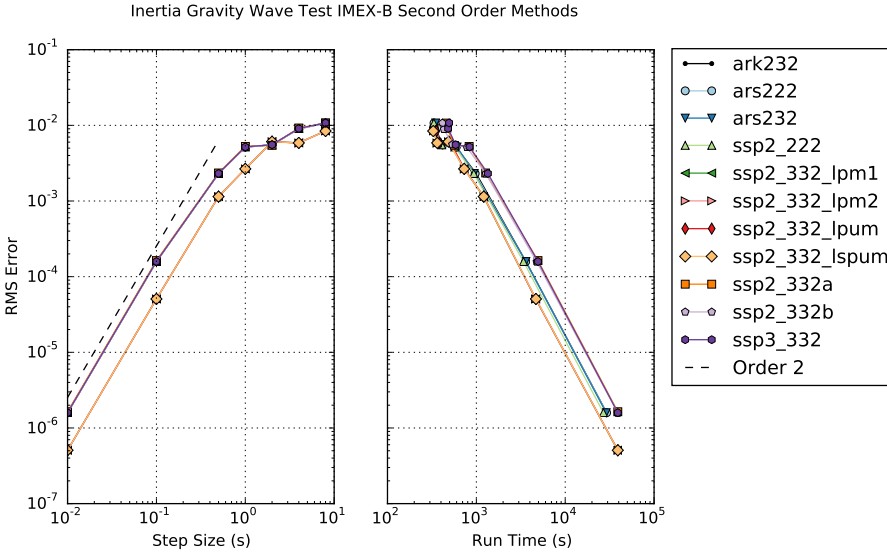

**Figure 5.** Accuracy (left) and efficiency (right) for the second order methods using the Rosenbrock-like approach with the IMEX-B splitting. The ARK methods fall into two groups with similar accuracy. The dashed line represents second order convergence. Results using multiple Newton iterations to compute the stage solutions give nearly identical error values to the Rosenbrock-like results but increase the computational cost by at least 10%.



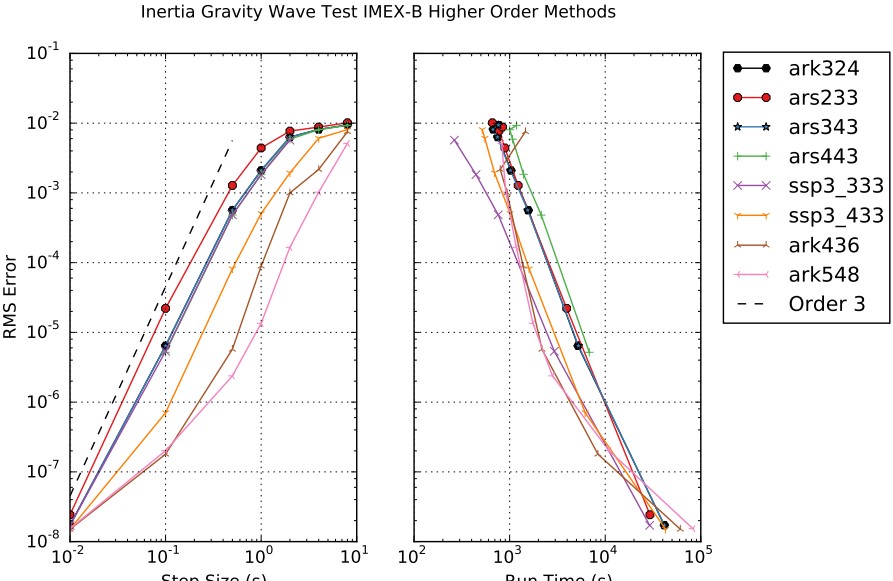

**Figure 6.** Accuracy (left) and efficiency (right) for the third, fourth, and fifth order methods using the Rosenbrock-like approach with the IMEX-B splitting. The dashed line represents third order convergence. Results using multiple Newton iterations to compute the stage solutions give nearly identical error values to the Rosenbrock-like results but increase the computational cost by at least 10%.

perturbed by random noise. For a state variable $x$, the perturbed initial value is $x = x + \max(\kappa|x|, \kappa)$ where $\kappa$ is a normally distributed random number with mean 0, standard deviation $\varepsilon \times 10^{11}$, and $\varepsilon$ is machine epsilon. The factor $10^{11}$ was selected to produce a max absolute difference (compared to the unperturbed explicit solution) in the vertical velocity after one day that is approximately an order of magnitude smaller than the max absolute difference observed with the ARS232 scheme using a step

size of 200s. The explicit simulations utilize a Rayleigh sponge layer to damp problematic acoustic transients as, unlike with the ARK methods, there is not an implicit mechanism for dissipating these modes. The sponge layer is $8\,\text{km}$ thick with a maximum strength of $0.5$ and is applied after the RK update, by way of a backward Euler step, to relax all prognostic fields to the initial state continuously through the depth of the layer. The explicit simulations are advanced in time with the KGU35 method in Tempest using $h_n = 2\text{s}$ which is approximately the CFL step size for the simulation. The absolute maximum vertical velocity

over the domain is computed at 1 day intervals for each test and a 99% confidence interval for the mean maximum vertical velocity is computed for each day using the t-distribution (Devore, 2008) to provide an upper bound on what is considered an acceptable solution. Figure 7 shows the 99% confidence interval for the maximum vertical velocity (the light red region) and the maximum vertical velocities for the HEVI–A splitting with ARS343 using various time step sizes. In the first few days of simulation, the velocities are slightly larger in the HEVI and IMEX formulations due to the presence of transients that

are damped out by the presence of a Rayleigh sponge in our explicit simulations. Nonetheless these transients are small and the vertical velocity is very similar to our reference range. The purple region is defined as 10% of the maximum deviation and the differences due to transients early in time fall within this region. To account for momentary large deviations from





the confidence interval, the maximum vertical velocity of a method should fall in the reference range. The predictability of the solution breaks down over the last 15 days, and so small, brief excursions outside of the reference range should not be considered anomalous. In the example in Figure 7, ARS343 is stable with step sizes up to 450s. However, the results with the Rosenbrock-like approach (solid lines) produce exceptionally large vertical velocities that decrease with step size and an

5 acceptable solution is produced once the step size is below 300s. The solution using multiple Newton iterations (dashed line) is able to more accurately solve the nonlinear stage systems and yields an acceptable solution with a step size of 450s. The maximum acceptable time step size for the different splittings and integration methods using this methodology for defining an acceptable solution are given in Table 1. The corresponding normalized run times for the step sizes given in Table 2 are listed in Table 3.

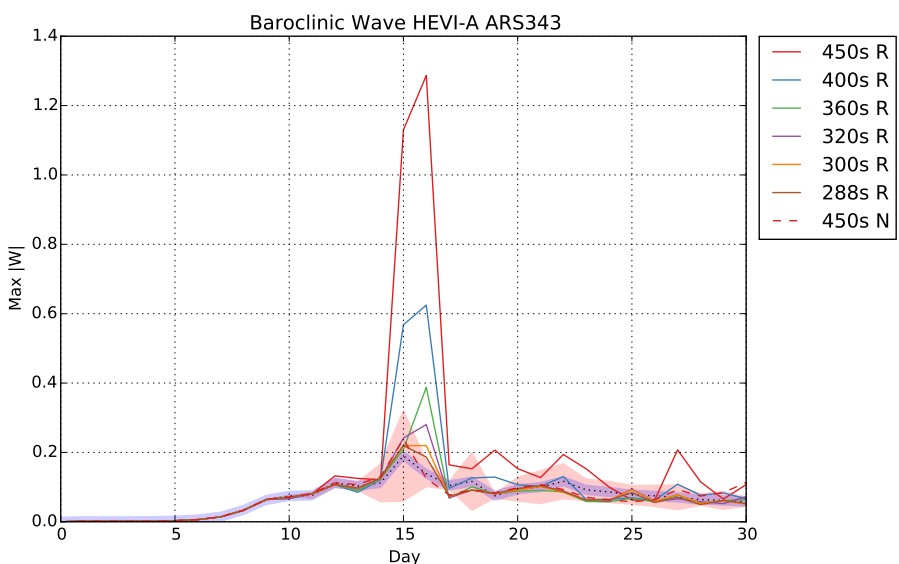

**Figure 7.** The maximum vertical velocity with ARS343 using the Rosenbrock-like approach (solid lines) and the Newton solver (dashed line) for various time step sizes. The light red region defines the 99% confidence interval from the explicit simulations with perturbed initial conditions, and the light purple region is 10% of the maximum deviation in the 99% confidence interval.

Unlike the gravity wave test, treating the thermodynamic advection explicitly (HEVI–C and D) reduces the maximum stable and acceptable step size for some of the integration schemes. As a result, the increased number of time steps with HEVI–C and D can lead to longer run times than with HEVI–A or B depending on the ARK method. Treating only the vertical velocity advection explicitly (HEVI–B) does not impact the maximum stable or acceptable step size, nor does it offer a significant advantage in run time over the HEVI–A setup. Handling more terms implicitly in IMEX–A and B can greatly increase the

maximum stable step size but, in general, this does not translate into faster run times due to the increased solve cost and the smaller step sizes need to produce a sufficiently accurate solution. However in a few cases with the Rosenbrock-like approach (ARK232, ARS222, ARS232, SSP2(332)lpm1, and SSP2(332)lpum2) IMEX–A runs are faster than results with HEVI–C and D because of the larger acceptable time step size with the IMEX–A splitting and the minimal increase in solver





cost due to the effectiveness of the vertical solve as a preconditioner (only 2 to 4 linear iterations are required per Newton iteration). The preconditioner is less effective in the IMEX–B splitting as more dynamics are included that are not treated by the preconditioner, so 16 to 25 linear iterations are needed per Newton iteration. As in the gravity wave test, the Newton solver does not preform well with the IMEX–B splitting and is unable to converge at step sizes for which the Rosenbrock-like

approach gives an acceptable answer.

In this test the increased accuracy and larger stability regions of the higher order methods enable bigger time step sizes than the second order methods and are somewhat less affected by the choice of HEVI splitting. The gains in step size are large enough to offset the third implicit stage solve required for ARK324 and ARS343, consistently making them the two fastest methods. The second order ARK232 and ARS232 methods give nearly identical performance and tie for the third best schemes.

When ARS233 is stable it is also an efficient approach but requires a Newton solver and is sensitive to the choice of splitting. The ARK324 adn ARS343 methods also highlight the potential advantage of the Newton solver over the Rosenbrock-like approach. With the exception of the SSP methods (discussed below), the second order methods studied produce acceptable solutions at their largest stable step size for HEVI splittings with either a Newton or Rosenbrock-like approach. As a result there is not a benefit from using the Newton solver for second order methods and the Rosenbrock-like approach is always more

efficient. At the larger stable step sizes enabled by higher order methods a Rosenbrock-like approach does not always give a sufficiently accurate answer, and a smaller step size is necessary to produce a good solution. Iterating to a converged stage value leads to better results at larger step sizes and, since only a few nonlinear iterations are necessary (on average 2 iterations per stage solve), a HEVI splitting with a Newton solver can outperform the Rosenbrock-like approach when the step size gain is sufficiently large.

While the other higher order schemes are also able to take larger time steps than the second order methods they require more function evaluations or implicit solves than ARK324 or ARS343, and the step size gains are not enough to overcome the additional costs. Two of the third order methods (ARS233 with the Rosenbrock-like solver and SSP3(333) with either solver) are not stable for 30 days at step sizes of at least 100s with any of the splittings. These failures are likely because the implicit parts are not L-stable (or even A-stable for SSP3(333)), and the fastest dynamics of the system are not sufficiently damped.

These methods did perform well in the gravity wave test case, which might have been due to the reduced domain size altering the eigenvalues of the system.

However, L-stability does not guarantee that a method will produce a good solution. All of the SSP methods tested, with the exception of SSP3(333), are L-stable but only SSP2(332)a gives acceptable results with the HEVI splittings. The other SSP methods are stable but give vertical velocities an order of magnitude larger than the mean solution value with step sizes

above 100s with the exception of SSP2(332)b which underestimates the vertical velocities. Acceptable solutions are produced with the IMEX–A and B splittings, suggesting that the inaccuracy with the SSP methods is related to the splitting error in the schemes. Comparing with the gravity wave results where the SSP methods were both accurate and efficient suggests that again the reduced domain size may have played a role in the quality of the results by altering the eigenvalues of the system, since the stability region of the explicit portion of many of the SSP methods does not include part of the imaginary axis.





**Table 1.** Approximate largest stable step size (seconds) for a 30 day run of the baroclinic wave test. Second order methods are shown in the top section of the table and higher order methods in the bottom section. As in the gravity wave tests the choice of a Rosenbrock-like or Newton solver does not impact the maximum stable step size except in the case of IMEX–B which fails to converge. While the methods are able to complete a 30 day run at the step sizes listed below, the solutions produced are not sufficiently accurate in all cases and depends on the solver choice. Table 2 shows the approximate largest step sizes that give acceptable solutions.

| Method | HEVI–A | HEVI–B | HEVI–C | HEVI–D | IMEX–A | IMEX–B |
|---|---|---|---|---|---|---|
| ARK232 | 200 | 200 | 135 | 135 | 480 | 480 |
| ARS222 | 160 | 160 | 100 ‖ $-^a$ | 100 ‖ $-^a$ | 360 | 360 |
| ARS232 | 200 | 200 | 135 | 135 | 480 | 540 |
| SSP2(222) | 160 | 160 | 160 | 160 | 360 | 540 |
| SSP2(332)lpm1 | 225 | 225 | $-^a$ | $-^a$ | 400 | 540 |
| SSP2(332)lpm2 | 225 | 225 | $-^a$ | $-^a$ | 400 | 540 |
| SSP2(332)lpum | 225 | 225 | $-^a$ | $-^a$ | 400 | 540 |
| SSP2(332)lspum | 300 | 300 | 108 ‖ 100 | 108 ‖ 100 | 450 | 540 |
| SSP2(332)a | 225 | 225 | 225 | 225 | 400 | 540 |
| SSP2(332)b | 225 | 225 | 225 | 225 | 400 | 540 |
| SSP3(332) | 320 | 320 | 320 | 320 | 540 | 540 |
| ARK324 | 400 | 400 | 400 | 400 | 300 | 540 |
| ARS233 | $-^a$ ‖ 320 | $-^a$ ‖ 320 | $-^a$ | $-^a$ | $-^a$ ‖ 180 | $-^a$ |
| ARS343 | 450 | 450 | 384 | 384 | 320 | 540 |
| ARS443 | 300 | 300 | 300 | 300 | 320 | 540 |
| SSP3(333) | $-^a$ | $-^a$ | $-^a$ | $-^a$ | $-^a$ | $-^a$ |
| SSP3(433) | 200 ‖ 216 | 200 ‖ 216 | 135 ‖ 150 | 135 ‖ 150 | 480 | 540 |
| ARK436 | 400 | 400 | $-^a$ | $-^a$ | 450 | 540 |
| ARK548 | 300 | 300 | $-^a$ | $-^a$ | 432 ‖ 450 | 432 |

a The method was not stable for 30 days with $h_n \geq 100$s

## 6 Conclusions

Considering the results of the gravity wave and baroclinic wave test cases the HEVI–A and B approaches are the most accurate and efficient of the implicit-explicit splittings considered. Treating some of the vertical dynamics of HEVI–A explicitly does not provide a noticeable gain in efficiency from simpler implicit systems and, in the case of HEVI–C and D, can lead to

5   reduced step sizes in the baroclinic wave test. Adding horizontally implicit terms to the HEVI–A formulation does increase the maximum stable step size, but the gains are not large enough to overcome the added cost of a globally implicit solve.

While SSP methods are the most accurate and efficient approaches in the gravity wave test case they do not preform well in the baroclinic wave test due to error from the choice of implicit-explicit splitting. The reduced domain size seemed to skew the gravity wave test results in favor of these methods while the ARK and ARS schemes perform well in both test cases.

10  Additionally, the gravity wave test case does not show a benefit, in terms of maximum stable step size, with higher order methods although it does highlight their greater efficiency when higher accuracy is required. Again, these results are likely due to the reduced domain size altering the eigenvalues of the system. In the baroclinic wave test on a full size Earth, higher order





**Table 2.** Approximate largest step size (in seconds) for a 30 day run of the baroclinic wave test that produces acceptable maximum vertical velocities over time. Second order methods are shown in the top section of the table and higher order methods in the bottom section. For entries separated by "||" the left value is the step size for the Rosenbrock-like approach and the right value is the step size for the Newton solver. When a single step size is given the Rosenbrock-like and Newton solvers gave acceptable solutions at the same step size.

| Method | HEVI–A | HEVI–B | HEVI–C | HEVI–D | IMEX–A | IMEX–B |
|---|---|---|---|---|---|---|
| ARK232 | 200 | 200 | 128 | 128 | 320 || 480 | 320 |
| ARS222 | 160 | 160 | 100 || $-^a$ | 100 || $-^a$ | 360 | 320 |
| ARS232 | 200 | 200 | 128 | 128 | 360 || 480 | 360 |
| SSP2(222) | $-^b$ | $-^b$ | $-^b$ | $-^b$ | $-^b$ | 360 |
| SSP2(332)lpm1 | $-^b$ | $-^b$ | $-^a$ | $-^a$ | 400 || 200 | 400 |
| SSP2(332)lpm2 | $-^b$ | $-^b$ | $-^a$ | $-^a$ | 100 || 150 | 450 |
| SSP2(332)lpum | $-^b$ | $-^b$ | $-^a$ | $-^a$ | 360 || – | 450 |
| SSP2(332)lspum | $-^b$ | $-^b$ | $-^b$ | $-^b$ | 120 || 450 | 450 |
| SSP2(332)a | 225 | 225 | 225 | 225 | 300 || 400 | 320 |
| SSP2(332)b | $-^b$ | $-^b$ | $-^b$ | $-^b$ | 270 || 400 | 320 |
| SSP3(332) | $-^b$ | $-^b$ | $-^b$ | $-^b$ | $-^b$ | 384 |
| ARK324 | 300 || 400 | 300 || 400 | 300 || 400 | 300 || 400 | 300 | 320 |
| ARS233 | $-^a$ || 320 | $-^a$ || 320 | $-^a$ | $-^a$ | $-^a$ ||180 | $-^a$ |
| ARS343 | 288 || 450 | 300 || 450 | 300 || 384 | 300 || 384 | 300 || 320 | 300 |
| ARS443 | 300 | 300 | 270 || 300 | 270 || 300 | 300 || 320 | 320 |
| SSP3(333) | $-^a$ | $-^a$ | $-^a$ | $-^a$ | $-^a$ | $-^a$ |
| SSP3(433) | $-^b$ | $-^b$ | $-^b$ | $-^b$ | 480 | 480 |
| ARK436 | 400 | 400 | $-^a$ | $-^a$ | 384 || 450 | 384 |
| ARK548 | 300 | 300 | $-^a$ | $-^a$ | 400 || 450 | 384 |

a The method was not stable for 30 days with $h_n \geq 100s$
b The method was unable to produce an acceptable solution with $h_n \geq 100s$

methods produce accurate solutions at step sizes large enough to have faster run times than second order schemes involving fewer implicit solves.

At the larger time step sizes enabled by higher order methods in the baroclinic wave tests the choice of nonlinear solver approach becomes an important consideration. A Rosenbrock-like approach limits the cost associated with multiple Newton iterations but may require a reduced step size to obtain an accurate solution. Taking larger steps is possible by iterating stage solutions to convergence with Newton's method. The additional cost is minimal and can be offset by the larger step size. The choice of predictor values was not considered in this work but could lead to more efficient nonlinear solves with Newton's method or more accurate Rosenbrock-like schemes.

The HEVI–A and B configurations produced nearly identical results while the HEVI–C and D options were problematic from some methods in the baroclinic wave test. Since HEVI–A employs the same discretization for Kinetic energy transport as vertical mass transport without a significant difference in computational cost it is preferred over the HEVI–B option. Overall, ARS343 is the best performing method and is followed closely by the third order ARK324 method. The second order ARS232 and ARK232 methods highlighted in Weller et al. (2013) and Giraldo et al. (2013) using the Rosenbrock-like were also very efficient options.





**Table 3.** The corresponding run times for the approximate largest acceptable step sizes in Table 2. Second order methods are shown in the top section of the table and higher order methods in the bottom section. The times have been normalized by the fastest simulation time, HEVI–B using ARS343 with the Newton solver (1,372.483s). The value on the left of the "‖" divider is the time for the Rosenbrock-like approach and the value on the right is the time for the Newton solver.

| Method | HEVI–A | HEVI–B | HEVI–C | HEVI–D | IMEX–A | IMEX–B |
|---|---|---|---|---|---|---|
| ARK232 | 1.19 ‖ 1.61 | 1.19 ‖ 1.61 | 1.81 ‖ 2.47 | 1.82 ‖ 2.43 | 1.77 ‖ 2.04 | 6.84 |
| ARS222 | 1.48 ‖ 2.01 | 1.48 ‖ 2.02 | 2.30 ‖ $-^a$ | 2.31 ‖ $-^a$ | 1.56 ‖ 2.31 | 6.67 |
| ARS232 | 1.19 ‖ 1.61 | 1.20 ‖ 1.60 | 1.83 ‖ 2.46 | 1.81 ‖ 2.43 | 1.53 ‖ 2.02 | 6.49 |
| SSP2(222) | $-^b$ | $-^b$ | $-^b$ | $-^b$ | $-^b$ | 6.47 |
| SSP2(332)lpm1 | $-^b$ | $-^b$ | $-^a$ | $-^a$ | 1.74 ‖ 4.76 | 6.86 |
| SSP2(332)lpm2 | $-^b$ | $-^b$ | $-^a$ | $-^a$ | 5.44 ‖ 6.01 | 6.62 |
| SSP2(332)lpum | $-^b$ | $-^b$ | $-^b$ | $-^b$ | 1.89 ‖ – | 6.60 |
| SSP2(332)lspum | $-^b$ | $-^b$ | $-^b$ | $-^b$ | 4.79 ‖ 2.84 | 6.69 |
| SSP2(332)a | 1.27 ‖ 1.83 | 1.25 ‖ 1.79 | 1.24 ‖ 1.92 | 1.24 ‖ 1.86 | 2.29 ‖ 3.02 | 9.48 |
| SSP2(332)b | $-^b$ | $-^b$ | $-^b$ | $-^b$ | 2.50 ‖ 3.01 | 8.90 |
| SSP3(332) | $-^b$ | $-^b$ | $-^b$ | $-^b$ | $-^b$ | 9.36 |
| ARK324 | 1.06 ‖ 1.14 | 1.07 ‖ 1.13 | 1.06 ‖ 1.23 | 1.05 ‖ 1.21 | 2.93 ‖ 4.41 | 13.28 |
| ARS233 | $-^a$ ‖ 1.03 | $-^a$ ‖ 1.02 | $-^a$ | $-^a$ | $-^a$ ‖ 4.38 | $-^a$ |
| ARS343 | 1.10 ‖ 1.02 | 1.09 ‖ **1.00** | 1.07 ‖ 1.21 | 1.05 ‖ 1.19 | 2.86 ‖ 4.08 | 13.21 |
| ARS443 | 1.34 ‖ 1.90 | 1.32 ‖ 1.90 | 1.46 ‖ 1.88 | 1.45 ‖ 1.84 | 3.82 ‖ 5.48 | 19.26 |
| SSP3(333) | $-^a$ | $-^a$ | $-^a$ | $-^a$ | $-^a$ | $-^a$ |
| SSP3(433) | $-^b$ | $-^b$ | $-^b$ | $-^b$ | 2.27 ‖ 3.97 | 10.19 |
| ARK436 | 1.21 ‖ 1.76 | 1.20 ‖ 1.74 | $-^a$ | $-^a$ | 3.28 ‖ 4.65 | 14.14 |
| ARK548 | 2.14 ‖ 3.16 | 2.15 ‖ 3.11 | $-^a$ | $-^a$ | 3.96 ‖ 5.75 | 17.31 |

a The method was not stable for 30 days with $h_n \geq 100$s
b The method was unable to produce an acceptable solution with $h_n \geq 100$s

The ARK324 and ARK232 are of particular interest as both include an embedded method which will be leveraged for future studies on temporal adaptivity in atmospheric simulations using ARKode. Varying the time step size can enable greater efficiency by placing temporal accuracy where is it needed most to capture dynamics of interest. Additionally, we plan on further evaluating the methods in this study on the 2016 dynamical core model intercomparision project (DCMIP2016) test

5      cases, to better understand the impacts of coupling with simplified physics on performance of implicit-explicit splittings and integration methods.

**Code and data availability**

Tempest is available through the Git repository https://github.com/paullric/tempestmodel, and ARKode is available as part of the SUNDIALS library of solvers downloadable from http://computation.llnl.gov/projects/sundials. The version of Tempest

10     that includes ARKode interfaces used for this work and splittings with horizontally implicit terms is available in the Git repository https://github.com/gardner48/tempestmodel.The versions of Tempest with ARKode interfaces used in this work are archived at https://doi.org/10.5281/zenodo.1043549.



## Appendix A: ARK Method Properties

In Table A1, we provide a variety of theoretical properties of each of the ARK methods used in this paper. While we do not reproduce each Butcher table here, references for each method are provided in Section 3.2. For each method, we provide the following information:

– number of implicit solves per step ($\boldsymbol{f}^I$ column) – the number of nonzero entries on the diagonal of $A^I$,

   – number of explicit stages ($\boldsymbol{f}^E$ column) – the total number of RK stages that involve calls to $\boldsymbol{f}^E$,

   – order – theoretical order of accuracy of the ERK method, the DIRK method and the overall ARK method (including coupling conditions),

   – stage order – theoretical order of accuracy of stages (relevant for order reduction on stiff problems), again for the ERK
stages, DIRK stages and the overall ARK stages,

   – stability – $A$, $L$, and $B$ stability for the DIRK portion of each method,

   – S.A. DIRK – if the DIRK method is stiffly-accurate (i.e., the last row of $A^I$ is the same as the $b^I$),

   – S.A. ERK – if the ERK method is "stiffly-accurate" (i.e., the last row of $A^E$ is the same as the $b^E$),

   – same solution weights ($\boldsymbol{b}$ column) – if the ERK and DIRK methods have the same weights to compute $\boldsymbol{y}_n$ (i.e., $\boldsymbol{b}^E = \boldsymbol{b}^I$),

– same abscissae ($\boldsymbol{c}$ column) – if the stages in the ERK and DIRK methods are evaluated at the same stage times (i.e.,
    $\boldsymbol{c}^E = \boldsymbol{c}^I$),

   – maximum stable explicit step along the imaginary axis – as this application has purely imaginary eigenvalues, we nu-
    merically compute the largest $y_{\max} \in \mathbb{R}$ such that the ERK portion of the method is stable for all $\lambda = iy$, $0 \le y \le y_{\max}$,
    using a bisection method with tolerance $10^{-6}$. If the method is analytically unstable for any nonzero values along the
imaginary axis, we list "0."

*Acknowledgements.* Support for this work was provided by the Department of Energy, Office of Science Scientific Discovery through Advanced Computing (SciDAC) project "A Non-hydrostatic Variable Resolution Atmospheric Model in ACME." This work was performed under the auspices of the U.S. Department of Energy by Lawrence Livermore National Laboratory under Contract DE-AC52-07NA27344. LLNL-JRNL-737448

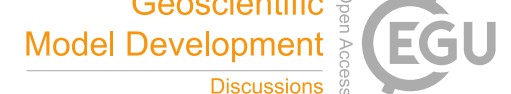


**Table A1.** Properties for each of the ARK methods used in this paper. The column headings are described in the above text.

| Method | $f^I$ | $f^E$ | Order | | | Stage Order | | | Stability | | | S.A. DIRK | S.A. ERK | $b$ | $c$ | Max Exp |
|---|---|---|---|---|---|---|---|---|---|---|---|---|---|---|---|---|
| | | | E | I | A | E | I | A | A | L | B | | | | | |
| ARK232 | 2 | 3 | 2 | 2 | 2 | 1 | 2 | 1 | ✓ | ✓ | ✗ | ✓ | ✗ | ✓ | ✓ | ∼1.73 |
| ARS222 | 2 | 3 | 2 | 2 | 2 | 1 | 1 | 1 | ✓ | ✓ | ✗ | ✓ | ✓ | ✗ | ✓ | 0 |
| ARS232 | 2 | 3 | 2 | 2 | 2 | 1 | 1 | 1 | ✓ | ✓ | ✗ | ✓ | ✗ | ✓ | ✓ | ∼1.73 |
| SSP2(222) | 2 | 2 | 2 | 2 | 2 | 1 | 1 | 0 | ✓ | ✓ | ✓ | ✗ | ✗ | ✓ | ✗ | 0 |
| SSP2(332)lpm1 | 3 | 3 | 2 | 2 | 2 | 1 | 1 | 0 | ✓ | ✓ | ✗ | ✗ | ✗ | ✓ | ✗ | 0 |
| SSP2(332)lpm2 | 3 | 3 | 2 | 2 | 2 | 1 | 1 | 0 | ✓ | ✓ | ✗ | ✗ | ✗ | ✓ | ✗ | 0 |
| SSP2(332)lpum | 3 | 3 | 2 | 2 | 2 | 1 | 1 | 0 | ✓ | ✓ | ✗ | ✗ | ✗ | ✓ | ✗ | 0 |
| SSP2(332)lspum | 3 | 3 | 2 | 2 | 2 | 1 | 1 | 0 | ✓ | ✓ | ✗ | ✗ | ✗ | ✓ | ✗ | ∼1.2 |
| SSP2(332)a | 3 | 3 | 2 | 2 | 2 | 1 | 1 | 0 | ✓ | ✓ | ✗ | ✓ | ✗ | ✓ | ✗ | 0 |
| SSP2(332)b | 3 | 3 | 2 | 2 | 2 | 1 | 1 | 0 | ✓ | ✓ | ✗ | ✓ | ✗ | ✓ | ✗ | 0 |
| SSP3(332) | 3 | 3 | 3 | 2 | 2 | 1 | 1 | 0 | ✓ | ✓ | ✗ | ✗ | ✗ | ✓ | ✗ | ∼1.73 |
| ARK324 | 3 | 4 | 3 | 3 | 3 | 1 | 2 | 1 | ✓ | ✓ | ✗ | ✓ | ✗ | ✓ | ✓ | ∼2.48 |
| ARS233 | 2 | 3 | 3 | 4 | 3 | 1 | 1 | 1 | ✓ | ✗ | ✓ | ✗ | ✗ | ✓ | ✓ | ∼1.73 |
| ARS343 | 3 | 4 | 3 | 3 | 3 | 1 | 1 | 1 | ✓ | ✓ | ✗ | ✓ | ✗ | ✓ | ✓ | ∼2.83 |
| ARS443 | 4 | 4 | 3 | 3 | 3 | 1 | 1 | 1 | ✓ | ✓ | ✗ | ✓ | ✓ | ✗ | ✓ | ∼1.57 |
| SSP3(333) | 2 | 3 | 3 | 3 | 3 | 1 | 1 | 1 | ✗ | ✗ | ✗ | ✓ | ✗ | ✓ | ✗ | ∼1.73 |
| SSP3(433) | 4 | 3 | 3 | 3 | 3 | 1 | 1 | 0 | ✓ | ✓ | ✗ | ✗ | ✗ | ✓ | ✗ | ∼1.73 |
| ARK436 | 5 | 6 | 4 | 4 | 4 | 1 | 2 | 1 | ✓ | ✓ | ✗ | ✓ | ✗ | ✓ | ✓ | ∼4.00 |
| ARK548 | 7 | 8 | 5 | 5 | 5 | 1 | 2 | 1 | ✓ | ✓ | ✗ | ✓ | ✗ | ✓ | ✓ | ∼0.79 |

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
