# Peer review of "Implicit-explicit (IMEX) Runge-Kutta methods for non-hydrostatic atmospheric models"

_Geoscientific Model Development, 2017_

## Referee Comment (RC1) · Anonymous Referee #1 · 29 Nov 2017

This paper is very thorough and clear. It also addresses the important issue of evaluating which IMEX Runge–Kutta methods are best suited for simulation of non-hydrostatic atmospheric models. Many methods are considered and the results are clearly reported as well as the reason these methods perform the way they do. I would be curious to see how this compares to the IMEX SSP methods presented in the recent paper by Conde et al, especially their third order method which is based on the explicit Shu-Osher method and a well-known family of implicit methods.

---

## Referee Comment (RC2) · Anonymous Referee #2 · 4 Dec 2017

This is a well done study in my opinion and I have very few suggestions.

1- In general I submit that conservation is important as well and might be a relevant criterion to be added to the list of properties. Not all methods are conservative: e.g., ARS222. Some discussion is in Giraldo et al. (2013). That imposes additional constraints on the conservative methods. Some methods in Weller et al. (2013) and Lock et al. (2014) may not be conservative.

2- Page 3 top; one-dimensional IMEX in NUMA is more or less equivalent to HEVI. Perhaps that can be stated.

3- Page 11 bottom (line 28): no communication only when the partition is done by vertical columns.

[Figure]

4- Page 20, lines ∼25-30: I'm surprised by the SSP performance, I am not sure that the coupling is the issue because most of them have the same coupling order. This could be speculative, but not a bad guess. However, I agree with the eigenvalue distribution argument.

―――――――――――――――――――――――

---

## Author Comment (AC1) · 20 Jan 2018

RC1: "*I would be curious to see how this compares to the IMEX SSP methods presented in the recent paper by Conde et al, especially their third order method which is based on the explicit Shu-Osher method and a well-known family of implicit methods.*"

The two third order IMEX SSP methods presented in Conde et al. (2017), that are based on the Shu-Osher method and paired with two members from a family of DIRK

methods, are given by the Butcher tableaux

$$
\begin{array}{c|ccc}
0 & 0 & 0 & 0 \\
1 & 1 & 0 & 0 \\
1/2 & 1/4 & 1/4 & 0 \\
\hline
& 1/6 & 1/6 & 2/3
\end{array}
\qquad
\begin{array}{c|ccc}
0 & 0 & 0 & 0 \\
1 & 0 & 1 & 0 \\
1/2 & 1/6 & -1/3 & 2/3 \\
\hline
& 1/6 & 1/6 & 2/3
\end{array}
\tag{1}
$$

and

$$
\begin{array}{c|ccc}
0 & 0 & 0 & 0 \\
1 & 1 & 0 & 0 \\
1/2 & 1/4 & 1/4 & 0 \\
\hline
& 1/6 & 1/6 & 2/3
\end{array}
\qquad
\begin{array}{c|ccc}
0 & 0 & 0 & 0 \\
1 & 4\gamma+2\beta & 1-4\gamma-2\beta & 0 \\
1/2 & 1/2-\beta-\gamma & \gamma & \beta \\
\hline
& 1/6 & 1/6 & 2/3
\end{array}
\tag{2}
$$

where $\beta = \sqrt{3}/6 + 1/2$ and $\gamma = -(\sqrt{3}+1)/8$. We will refer to methods (1) and (2) as SSP3(333)b and SSP3(333)c respectively to distinguish these methods from the third order method in Higueras (2009), which we will refer to as SSP3(333)a and that uses the same explicit method. The second order SSP3(332) and third order SSP3(433) methods from Pareschi and Russo (2005) also utilize the same explicit tableau. Therefore any differences in performance will stem from the choice of implicit method. SSP3(333)b and c are A-stable, which is an improvement in linear stability compared to SSP3(333)a, but SSP3(333)b and c are not L-stable like SSP3(332) and SSP3(433). Based on the similarities to methods examined in the paper we would expect the performance of SSP3(333)b and c to be between that of the Higueras (2009) and Pareschi and Russo (2005) methods.

However, in preparing this response we tested these methods and found their performance to be better than anticipated, depending on the choice of splitting and nonlinear solver approach. Their behavior is similar to ARS233 which is also a third order method with two implicit solves. We summarize the results for these runs below and will add the results for these methods the paper and update the discussion accordingly.

Testing SSP3(333)b and c on the baroclinic wave test case we find that both methods, like SSP3(333)a and ARS233, are unable to complete a 30 day simulation for any of the splitting options when using the Rosenbrock-like solver approach using step sizes of 100s or larger. However, with the Newton solver SSP3(333)b and c have the same approximate maximum stable step size as SSP3(332) and ARS233, 320s, for the HEVI–A and B splittings. Also like ARS233, SSP3(333)b and c fail to complete 30 day simulations with the HEVI–C, HEVI–D, and IMEX–B splittings and do not perform as well with the IMEX–A option. Unlike the majority of the other SSP methods, SSP3(333)b and c produce acceptably accurate results for HEVI–A and B when using the maximum time step size.

Compared to the other SSP methods considered, SSP3(333)b and c are the only methods with the same **c** values for both the implicit and explicit methods. While having identical **c** values is not necessary for producing acceptable solutions (e.g. SSP2(332)a), having the stage values aligned in time appears beneficial. Because SSP3(333)b and c have a relatively large acceptable step and require only two implicit solves per step, the methods are very efficient with normalized run times of 1.04 and 1.03 respectively with the HEVI–A splitting and 1.02 for both methods with HEVI–B. While these methods have the second fastest time, just behind ARS343 and ahead of ARK324, they are less robust to the choice of splitting and nonlinear solver than ARS343 and ARK324.

RC2: "*1- In general I submit that conservation is important as well and might be a relevant criterion to be added to the list of properties. Not all methods are conservative: e.g., ARS222. Some discussion is in Giraldo et al. (2013). That imposes additional constraints on the conservative methods. Some methods in Weller et al. (2013) and Lock et al. (2014) may not be conservative.*"

We agree that the conservation properties of a method might be a relevant criterion to

consider as well. Although we did not explicitly state the conservation properties of the ARK methods tested, Table 4 includes a column indicating which ARK methods have the same implicit and explicit **b** values. ARK methods with this property will preserve linear invariants, and, of the ARK methods tested, only ARS222 and ARS443 do not have this property. We will add a comment on conservation when presenting the methods and note this in the text describing the properties summarized in Table 4.

RC2: "*2- Page 3 top; one-dimensional IMEX in NUMA is more or less equivalent to HEVI. Perhaps that can be stated.*"

Yes, the one-dimensional IMEX splitting in Giraldo et al. (2013) is a HEVI splitting and we will note this in the paper.

RC2: "*3- Page 11 bottom (line 28): no communication only when the partition is done by vertical columns.*"

Thank you for the suggestion, we will clarify the text to note that parallel communication is not necessary when the domain is decomposed into vertical columns.

RC2: "*4- Page 20, lines~25-30: I'm surprised by the SSP performance, I am not sure that the coupling is the issue because most of them have the same coupling order. This could be speculative, but not a bad guess. However, I agree with the eigenvalue distribution argument.*"

Based on the performance difference between the splittings we suspect coupling error is playing a role but it is likely not the sole factor. Tests with two SSP IMEX methods

from Conde et al. (2017), as suggested in referee #1's comments and discussed above, show much better performance in the baroclinic wave test case with the HEVI–A and B splittings. Results and discussion of the Conde et al. (2017) methods will be added to the paper.

---

## Editor Comment (EC1) · S. Marras (Editor) · 22 Jan 2018

Dear Authors,

Based on you the reviews received by both reviewers and you comments and corrections to the manuscript, I am happy to accept your paper "Implicit-explicit (IMEX) Runge-Kutta methods for non-hydrostatic atmospheric models" for publication in Geoscientific Model Development.

Best regards, Simone Marras Topical Editor
* * *

---

## Author Response (AR1)

RC1: "*I would be curious to see how this compares to the IMEX SSP methods presented in the recent paper by Conde et al, especially their third order method which is based on the explicit Shu-Osher method and a well-known family of implicit methods.*"

As discussed in the Author Response to Referee Comments (AC1) we have evaluated the two third order IMEX SSP methods presented in Conde et al. (2017) based on the Shu-Osher method and found that their performance is similar to ARS233 which is also a third order method with two implicit solves.

In the revised manuscript we have updated the Additive Runge-Kutta Methods subsection to include these two methods (referred to as SSP3(333)b and SSP3(333)c) and noted they are the only SSP methods tested that have the same stage times for the implicit and explicit methods. The corresponding table of method properties (Table A1) has also been updated to include these methods. In the Gravity Wave and Baroclinic Wave subsections, the discussion of numerical results and the related figures and tables have also been updated to include the SSP3(333)b and SSP3(333)c methods. Finally, the Conclusions section has been updated to mention the performance of the ARS233, SSP3(333)b, and SSP3(333)c methods.

RC2: "*1- In general I submit that conservation is important as well and might be a relevant criterion to be added to the list of properties. Not all methods are conservative: e.g., ARS222. Some discussion is in Giraldo et al. (2013). That imposes additional constraints on the conservative methods. Some methods in Weller et al. (2013) and Lock et al. (2014) may not be conservative.*"

We agree that the conservation properties of a method might be a relevant criterion as well and have updated the Additive Runge-Kutta subsection to note that all methods considered, except ARS222 and ARS443, have this property. The description of the table of method properties (Table A1) in the appendix has also been updated to make clear which methods preserve linear invariants.

RC2: "*2- Page 3 top; one-dimensional IMEX in NUMA is more or less equivalent to HEVI. Perhaps that can be stated.*"

The one-dimensional IMEX splitting in Giraldo et al. (2013) is a HEVI splitting and we have updated the text in the Introduction section to state this.

RC2: "*3- Page 11 bottom (line 28): no communication only when the partition is done by vertical columns.*"

This is correct and we have revised the text in the Linear Solvers section to clarify that parallel communication is not necessary since the domain is decomposed by vertical columns.

RC2: "*4- Page 20, lines~25-30: I'm surprised by the SSP performance, I am not sure that the coupling is the issue because most of them have the same coupling order. This could be speculative, but not a bad guess. However, I agree with the eigenvalue distribution argument.*"

We suspect coupling error is playing a role but it is likely not the sole factor. As such we have revised the text in the Baroclinic Wave subsection to clarify that splitting error may be related to the performance of the SSP methods. Additionally, the revised Baroclinic Wave subsection includes the results of SSP3(333)b and SSP3(333)c which show much better performance with the HEVI–A and B splittings.

[revised manuscript text omitted]